# Microphthalmia-Associated Transcription Factor-Dependent Melanoma Cell Adhesion Molecule Activation Promotes Peritoneal Metastasis of Ovarian Cancer

**DOI:** 10.3390/ijms21249776

**Published:** 2020-12-21

**Authors:** Kazuhisa Kitami, Masato Yoshihara, Yoshihiro Koya, Mai Sugiyama, Shohei Iyoshi, Kaname Uno, Kazumasa Mogi, Sho Tano, Hiroki Fujimoto, Akihiro Nawa, Fumitaka Kikkawa, Hiroaki Kajiyama

**Affiliations:** 1Department of Obstetrics and Gynecology, Nagoya University Graduate School of Medicine, 65 Tsuruma-cho, Showa-ku, Nagoya 466-8550, Japan; kitami.kazuhisa@med.nagoya-u.ac.jp (K.K.); iyoshi.shohei@med.nagoya-u.ac.jp (S.I.); uno.kaname@med.nagoya-u.ac.jp (K.U.); mogi.kazumasa@med.nagoya-u.ac.jp (K.M.); tano.sho@med.nagoya-u.ac.jp (S.T.); fujimoto.hiroki@med.nagoya-u.ac.jp (H.F.); kikkawaf@med.nagoya-u.ac.jp (F.K.); kajiyama@med.nagoya-u.ac.jp (H.K.); 2Bell Research Center, Department of Obstetrics and Gynecology Collaborative Research, Nagoya University Graduate School of Medicine, 65 Tsuruma-cho, Showa-ku, Nagoya 466-8550, Japan; mai-sugiyama@kishokai.or.jp (M.S.); nawa2005@med.nagoya-u.ac.jp (A.N.); 3Spemann Graduate School of Biology and Medicine, University of Freiburg, Albertstr. 19A, 79104 Freiburg, Germany; 4Faculty of Medicine, Lund University, Sölvegatan 19, 22184 Lund, Sweden

**Keywords:** MITF, MCAM, CD146, ovarian cancer, metastasis, invasion, migration, mesothelial, EMT

## Abstract

Ovarian cancer (OvCa) is one of the leading causes of death due to its high metastasis rate to the peritoneum. Recurrent peritoneal tumors also develop despite the use of conventional platinum-based chemotherapies. Therefore, it is still important to explore the factors associated with peritoneal metastasis, as these predict the prognosis of patients with OvCa. In this study, we investigated the function of microphthalmia-associated transcription factor (MITF), which contributes to the development of melanoma, in epithelial ovarian cancer (OvCa). High MITF expression was significantly associated with a poor prognosis in OvCa. Notably, MITF contributed to the motility and invasion of OvCa cells, and specifically with their peri-mesothelial migration. In addition, MITF-positive cells expressed the melanoma cell adhesion molecule (MCAM/CD146), which was initially identified as a marker of melanoma progression and metastasis, and MCAM expression was regulated by MITF. MCAM was also identified as a significant prognostic factor for poor progression-free survival in patients with OvCa. Collectively, our results suggest that MITF is a novel therapeutic target that potentially promotes peritoneal metastasis of OvCa.

## 1. Introduction

More than half of all patients with ovarian cancer (OvCa) are diagnosed at an advanced stage, contributing to the fact that OvCa is the leading cause of death in the field of gynecology [1,2,3]. Peritoneal dissemination, which is one of the most common types of metastasis in the abdominal cavity, is frequently observed in patients with advanced OvCa [4,5]. Primary peritoneal carcinoma and advanced OvCa are similar in having peritoneal dissemination and are treated similarly. The recommended treatment integrates aggressive cytoreductive surgery and systemic chemotherapy to remove the macroscopic tumor, eradicate the microscopic residual disease, and control the microscopic metastasis [6]. Even when primary tumors are optimally resected, recurrent tumors frequently emerge in the peritoneum despite the use of conventional platinum-based chemotherapies [7,8] and the estimated median progression-free survival (PFS) is approximately 12–18 months [9]. Therefore, it remains important to explore the factors associated with the peritoneal metastasis of OvCa, as these can serve not only as biomarkers that predict patient prognosis, but also as therapeutic targets in refractory OvCa.

The microphthalmia-associated transcription factor (MITF) is a transcription factor of the basic helix-loop-helix–leucine zipper type that is essential for propagation of the melanocyte lineage. MITF is a melanoma oncogene and plays a central role in melanoma progression [10]. Furthermore, MITF is involved in cell cycle regulation [11,12], cell survival [13,14], cell differentiation [15], and cell migration [16,17]. In addition to melanoma, several tumors are associated with the dysregulation of the MITF gene, including renal cell carcinoma [18], clear cell sarcoma [19], prostate cancer [20], and chronic myeloid leukemia [21], but there are no reports of its involvement in OvCa.

We hypothesized that, in addition to its role in melanoma, MITF is associated with the progression of OvCa. Since the spread of cancer cells to the peritoneum is the most important step in the development of peritoneal dissemination, cell motility and invasion are expected to play a central role in OvCa metastasis. To verify this hypothesis, we evaluated the expression level of MITF in primary OvCa tissues and its association with prognosis. The roles of MITF and its targets in the motility and invasive abilities of OvCa cells also were assessed, using in vitro experimental models. Our results suggest that MITF modulates the melanoma cell adhesion molecule (MCAM/CD146), which in turn promotes peritoneal dissemination. As such, MITF may be a novel molecular target that potentially predicts prognosis in patients with OvCa.

## 2. Results

### 2.1. High MITF Expression Was Related to Poor Prognosis of Patients with OvCa

To determine whether MITF expression in tumor cells influenced the progression of OvCa, we performed IHC analysis of MITF in primary OvCa tissue samples. Representative IHC images from patients with advanced OvCa showed elevated MITF expression in tumor cells, while low MITF expression was observed in patients with early-stage OvCa (Figure 1A,B). We also analyzed MITF expression in primary OvCa tissue samples (six samples) by immunoblotting. MITF expression was found in all samples, and expression levels varied according to tumor characteristics, including clinical stage (Figure 1C, Appendix A). MITF was previously shown to be associated with prognosis in patients with melanoma [22] and renal cell carcinoma [23]. As expected, then, the OvCa expression database suggested that tumors with a high expression of MITF were associated with significantly worse prognosis than those with low expression, in all patients and in those with stage III disease (Figure 1D). Collectively, high MITF expression was associated with shorter progression-free survival in patients with OvCa, indicating that elevation of MITF in tumor cells is linked to metastatic progression of OvCa.

### 2.2. MITF Was Expressed in Mesenchymal-Like OvCa Cells with High Invasive and Migratory Potential

We examined the MITF expression level in representative OvCa cell lines with immunofluorescence and immunoblotting. Results demonstrated that ES-2, NOE, and NOS1 cells showed high MITF expression, whereas NOS2, NOS3, and SKOV3 showed low MITF expression (Figure 2A,B). To investigate the association between MITF expression and cellular morphology, we evaluated the shape of each type of OvCa cell and calculated the cell aspect ratio as an indicator of mesenchymal structure. ES-2 cells demonstrated a higher cell aspect ratio than that of NOS2 cells (Figure 2C). There was a positive correlation between MITF expression levels and cell aspect ratios, but the association was not significant (Figure 2D). We also analyzed epithelial–mesenchymal transition (EMT) markers in OvCa cell lines. E-cadherin was used as an epithelial marker, and N-cadherin and vimentin were used as mesenchymal markers (Figure 2B). The results suggested that ES-2, NOE, HEY, SKOV-3, and A2780 cells had mesenchymal properties, whereas CAOV3, OVCAR3, NOS1, NOS2, and NOS3 cells had epithelial properties. The results of the immunoblot analysis also were correlated with the aspect ratio. Furthermore, the mesenchymal-like OvCa cells ES-2, NOE, and HEY had high invasive and migratory potential in vitro and high metastatic potential in vivo [24]. Therefore, it was supposed that MITF may contribute to the high metastatic potential of progressive OvCa (Figure 2B–D).

Since MITF contributes to high invasive and migratory potential in melanoma and renal carcinoma [25,26,27], we examined whether the same was true in OvCa (Figure 2E–H). First, we knocked down MITF in ES-2 and NOE cell lines, since both demonstrate high invasiveness and motility as well as higher MITF expression than the other cell lines in this study. With semi-quantitative PCR, we confirmed that siRNA targeting effectively downregulated MITF in both ES-2 and NOE cells (Figure 2F). Then, we investigated cell migration and invasion with a transwell chamber system, with no coating to examine migration and with a Matrigel coating to evaluate invasion. In both cell lines, MITF downregulation significantly decreased the migration and invasion compared to the control condition (Figure 2G,H), whereas it did not decrease the migration and invasion in SKOV-3, which demonstrate low invasiveness and low MITF expression (Appendix A). Furthermore, A2780, NOS2, and NOS3, which have low MITF expression, have no migratory and invasive potential in vitro (Appendix A). Altogether, these data suggested that MITF is associated with the migration and invasion properties of invasive OvCa cells.

### 2.3. MITF Promoted Peri-Mesothelial Migration of OvCa Cells

OvCa cells that migrate from the ovary are known to metastasize to the peritoneum via ascites in the peritoneal cavity [28], and the upper surface of the peritoneum is covered by a monolayer of mesothelial cells [29]. Therefore, migration of OvCa cells along this monolayer may play an important role in the development of peritoneal metastasis [30]. We assessed this migration by mimicking the first stage of OvCa peritoneal metastasis using human peritoneal mesothelial cell (HPMC) monolayers on collagen-coated plates (Figure 3A,B). The GFP-labeled ES-2 cells transfected with control siRNA or siRNA against MITF were seeded on HPMCs, and the migration of cancer cells on HPMCs was continuously observed via time-lapse images (Figure 3C). ES-2 cells with MITF knockdown by siRNA showed significant decreases in migration distance (Figure 3D,E) as well as estimated migration velocity and mean-square displacement (Figure 3F,G). Overall, these results suggested that MITF promotes peri-mesothelial migration of OvCa cells, which potentially contributes to the poor prognosis of OvCa with high MITF expression.

### 2.4. Proteomic Analysis Revealed the Association of MITF with Cancer Signaling Pathways in OvCa Cells

To analyze the effect of MITF expression in OvCa cells, we performed tandem mass spectrometry in ES-2, NOE, and HEY cells treated with control or MITF-targeted siRNA. Evaluation of the obtained data with a principal component analysis map and clustered heat map suggested that each group shared alterations of protein expression (Figure 4A,B). Screening statistics indicated that 118 of 3016 detected proteins were altered (Figure 4C). Multiple pathway analysis revealed cancer-related signals, including MAPK (Figure 4D–F). Thus, proteomic analysis revealed that MITF was associated with cancer signaling pathways in OvCa cells.

### 2.5. MITF Regulated the Migration and Invasion Properties of OvCa Cells through MCAM

Next, we investigated whether MITF knockdown led to changes in the expression levels of MCAM, a cell-surface glycoprotein that regulates cell migration and is composed of five immunoglobulin-like domains [31]. MITF is reported to directly bind to the promoter region of MCAM and modulates its function [32]. Importantly, MCAM is associated with poor prognosis in various cancers [33], including ovarian cancer [34], and with enhanced motility of melanoblast and breast cancer cells [35]. Thus, MITF may regulate MCAM and promote OvCa metastasis by increasing cell motility and invasion. First, we used flow cytometry to evaluate MCAM expression on the surface of OvCa cell lines. The results showed that OvCa cells presented MCAM to various extents; there were high levels of MCAM expression in ES-2 and NOE cells, and low levels in SKOV-3 and NOS2 cells (Figure 5A). MCAM expression levels were also correlated with the cell aspect ratios of the different OvCa cell lines (Figure 5B). To confirm whether MCAM was regulated by MITF, we examined MCAM expression using MITF-targeted siRNA in ES-2 and NOE cells. The data indicated that siRNA effectively downregulated the expression of MCAM mRNA and protein in both cell types (Figure 5C,D). Collectively, these results suggested that MITF may regulate the expression of MCAM in OvCa cells.

Finally, we evaluated whether MCAM on OvCa cells affected migration and invasion, to determine if MITF regulates these properties through MCAM. ES-2 and NOE cells were treated with control or MCAM-targeted siRNA and plated on the transwell chamber as described (Figure 5E). Real-time PCR (RT-PCR) showed that MCAM was effectively knocked down in ES-2 and NOE cells (Figure 5F). As expected, MCAM downregulation significantly decreased motility and invasion compared to the control in both cell lines (Figure 5G,H). The OvCa expression database also clearly showed that tumors with high MCAM expression were associated with significantly shorter progression-free survival than those with low MCAM expression in all patients and in those with stage III disease (Figure 5I). In summary, our data suggest that MITF regulates MCAM expression and increases OvCa cell motility and invasiveness via MCAM, promoting peri-mesothelial migration and OvCa progression in OvCa cells (graphical abstract shown in Figure 5J).

## 3. Discussion

The primary finding of our study was that MITF increased tumor cell motility and invasion by modulating MCAM, thus promoting the progression of OvCa. This is the first report describing the significance of MITF in OvCa from the viewpoints of both clinical medicine and basic science. Our results suggest that MITF may be a novel molecular target that can potentially predict the prognosis of patients with OvCa.

In metastatic melanoma, MITF amplification was associated with a survival decrease of 5 years [36]. In this study, we first evaluated whether MITF demonstrated clinical significance in OvCa. The primary lesion of OvCa patients showed varying levels of MITF expression according to IHC analysis (Figure 1A). Immunoblotting analysis of high-grade serous carcinoma, the most common histological type of OvCa that accounts for approximately 70% of cases, found that all samples expressed MITF (Figure 1C), although the expression was variable. In addition, public data analysis (Figure 1D) clearly showed that high MITF expression was associated with poor prognosis regarding overall survival of patients with OvCa. This key finding indicates that MITF expression plays an important role in accelerating the progression of OvCa, for instance, by causing the progression of peritoneal dissemination. It is also suggested that positive MITF expression may be a useful biomarker for predicting poor prognosis in patients with OvCa.

It is hypothesized that carcinoma cells lose their epithelial characteristics and acquire certain mesenchymal properties that promote extracellular matrix invasion and distant metastasis in a process similar to that of epithelial–mesenchymal transition (EMT) [37,38]. Expression of EMT transcription factors (e.g., TWIST, SNAIL, SLAG, ZEB1, and ZEB2), which favor migration, invasion, and metastasis, causes cancer cells to switch from an epithelial to a mesenchymal phenotype [39]. After EMT, mesenchymal-like cancer cells show high motility and invasion properties [40]. Interestingly, a review showed that MITF plays an important role in migration and invasion in melanoma cells [41]. There have been no reports thus far on whether MITF contributes to cell migration and invasion in gynecological tumors, including ovarian cancer. Here we showed for the first time that MITF contributes strongly to migration and invasion abilities in OvCa cells (ES-2 and NOE cells), which exhibit mesenchymal-like features (Figure 2). The migration and invasion abilities of ES-2 and NOE cells decreased after transfection with siRNA against MITF (Figure 2G,H), suggesting that MITF plays an important role in migration and invasion abilities in OvCa.

OvCa is believed to cause peritoneal dissemination through microenvironmental cell-to-cell communication between the tumor and mesothelium, leading to increased progressive and metastatic potential [42,43]. Here, we analyzed the effect of MITF on the ability of OvCa cells to migrate on mesothelial cells. Downregulation of MITF in OvCa cells attenuated their migration ability on HPMC monolayers that mimicked the peritoneal surface (Figure 3C–G). Altogether, MITF is considered to regulate OvCa cell migration and increase the rate of peritoneal invasion and OvCa progression. As a consequence, MITF is expected to become a novel indicator that may predict the prognosis of OvCa patients.

MCAM is also called CD146, S-endo 1, MelCAM, and MUC18. It is an integral membrane cell adhesion molecule in the immunoglobulin-like gene superfamily and has an immunoglobulin-like extracellular domain and a cytoplasmic tail. A review showed that MCAM plays an important role in migration and invasion in melanoma cells [32,44]. MCAM also has been reported to contribute to cancer cell migration and invasion capacity in other solid tumors, including ovarian cancer [45,46,47]. However, no studies have investigated how MCAM expression is regulated in ovarian cancer cells. Here we found that ES-2 and NOE cells, which express MITF, also expressed MCAM on the cell surface (Figure 5A). We assessed MCAM expression, assuming that it was under the control of MITF. Treatment of siRNA against MITF reduced the MCAM mRNA levels as well as MCAM expression on the cell surface (Figure 5C,D), suggesting that MITF directly or indirectly regulates MCAM expression. We also showed that MCAM plays an important role in migration and invasion in vitro (Figure 5G,H). Additionally, high MCAM expression increased disease progression in OvCa patients, suggesting that MCAM modulation by MITF is a clinically important factor that accelerates OvCa progression, by promoting metastasis to the peritoneum, and worsens the prognosis. Therefore, MCAM also have the potential to be a biomarker to predict the prognosis of OvCa (Figure 5I) [48]. Collectively, the MITF/MCAM axis may significantly contribute to migration, invasion, and tumorigenicity in vivo, but research studies have not yet explored this question. However, there have been reports of inhibition of cancer metastasis, including in vivo, using MITF inhibitor [49], MCAM antibodies [50,51], and miRNA targeting MCAM [52,53]; it is hoped that inhibition of the MITF/MCAM axis will inhibit metastasis of ovarian cancer.

The strength of this study was that it evaluated the pro-tumoral functions of MITF using various clinical samples, including primary surgical specimens of OvCa as well as HPMCs extracted from resected omentum. The results obtained with these samples seemed to reflect the behavior of cells in living organisms. One main limitation, however, is that the findings only focused on the direct association between MITF and MCAM regarding the motility and invasion of OvCa cells, and other factors may also affect these functions. We believe that the key finding of this study, namely that MCAM modulation by MITF plays an important role in OvCa, provides a basis for future research.

In conclusion, MITF promoted the progression of OvCa by modulating MCAM expression and accelerating cell motility and invasion. It is necessary to characterize the clinical role of MITF in greater detail so that it can be used as a prognostic biomarker and a potential therapeutic target in OvCa.

## 4. Materials and Methods

### 4.1. Patient Data and Samples

Clinical data and tissue samples were collected from patients diagnosed with OvCa who underwent surgery at Nagoya University Hospital. All samples were resected from apparent tumor tissue and stored at −80 °C. Informed consent was obtained from patients according to the regulations set out by the Ethics Committee of Nagoya University (approved number: 2017–0497, approved date: 13 March 2018).

### 4.2. Immunohistochemistry

Immunohistochemistry of OvCa tissues obtained as described above was performed with a mouse monoclonal antibody against MITF. The histological type was specified according to the criteria of the World Health Organization classification. Slides were counterstained with Mayer’s hematoxylin (Wako, Osaka, Japan). Negative controls were run on all sections in blocking buffer, generated against unrelated antigens. The intensity of MITF immunostaining was scored as follows: 0 (negative), 1 (weak), 2 (medium), and 3 (strong). Tumors with a final staining score of 0–1 or 2–3 were defined as having low or high MITF expression, respectively. The scoring procedure was performed twice by two independent observers; each was blinded to the other’s scores and neither had any knowledge of the clinical parameters or other prognostic factors. The interobserver concordance rate was over 95%.

### 4.3. Public Data Analysis

The associations between the prognosis of OvCa patients and both MITF and MCAM mRNA expression were analyzed using a microarray gene expression database with a Kaplan–Meier plotter (http://kmplot.com/analysis/index.php?p=service&cancer=ovar).

### 4.4. Cell Lines

Established human ovarian cancer cell lines ES-2, NOE, HEY, SKOV3, A2780, CAOV3, OVCAR3, NOS1, NOS2, and NOS3 were grown in Dulbecco’s Modified Eagle’s Medium (DMEM: SIGMA, Tokyo, Japan) supplemented with heat-inactivated fetal bovine serum (FBS: Thermo Fisher Scientific, Yokohama, Japan) and 1% penicillin-streptomycin (Nacalai Tesque, Kyoto, Japan). All cells were cultured at 37 °C in a 5% CO_2_ humidified incubator. A2780 was purchased from the European Collection of Authenticated Cell Cultures (Salisbury, UK). HEY was purchased from Cosmo Bio (Tokyo, Japan). ES-2, OVCAR3, and SKOV3 cells were obtained from the American Type Culture Collection (Manassas, VA, USA). NOE, NOS1, NOS2, and NOS3 cells were established at the Department of Obstetrics and Gynecology, Nagoya University Graduate School of Medicine [54,55]. GFP-labeled ES-2 was established as described previously [56].

### 4.5. Immunofluorescence

Cells were incubated on glass coverslips. After demonstrating appropriate attachment, the cells were fixed using 4% paraformaldehyde in phosphate-buffered saline (PBS). They were then washed with PBS and treated with blocking buffer (5% goat serum and 0.3% Triton-X in PBS) at room temperature for 1 h and incubated overnight with anti-MITF antibody in antibody dilution buffer (1% bovine serum albumin (BSA) and 0.3% Triton-X in PBS) at 4 °C. The coverslips were washed three times with PBS and incubated with Alexa Fluor 488-conjugated secondary antibody (Jackson Immuno Research, West Grove, PA, USA), phalloidin-594, and 4′,6-diamidino-2-phenylindole (DAPI) for 1 h. After thorough washing, the cells on coverslips were overlaid with mounting medium (Cosmo Bio), and then analyzed with an FV1000 (Olympus, Tokyo, Japan).

### 4.6. Immunoblot Analysis

Immunoblot analysis was performed to detect MITF antigens in clinical samples and cell lines, as described previously [57]. The antibodies (Abs) utilized were anti-MITF, C5 (Santa Cruz Biotechnology, Dallas, TX, USA), E-cadherin (CST, Tokyo, Japan), N-cadherin (BD Biosciences, Tokyo, Japan), vimentin (Santa Cruz Biotechnology), and anti-GAPDH (CST).

### 4.7. Morphologic Analysis

After seeding ovarian cancer cell lines, up to five areas were randomly selected before confluency, and phase-contrast images were obtained with a digital microscope (BZ-X800, KEYENCE, Osaka, Japan). The aspect ratio of each cancer cell was calculated as the length of its longest axis divided by the length of its perpendicular axis, using the BZX-800 View Analyzer Software (KEYENCE).

### 4.8. Transwell Migration and Invasion Assays

The migration assay was performed in a transwell chamber (8 μm; Corning Japan, Tokyo, Japan). The invasion assay used a BioCoat Matrigel Invasion Chamber (8 μm; Corning). Cells were treated with the desired sequences of small interfering RNA (siRNAs). Briefly, cell suspensions (5 × 10^4^ cells/mL) were placed on top of the upper chamber (500 mL/well). During the 22-h incubation period, the cells moved through the membrane or matrix, and adhered to the bottom membrane of the insert. Motile cells were fixed with methanol and then stained with May–Grunwald–Giemsa. Images were obtained using an Olympus upright microscope and analyzed using ImageJ software (Java v1.52a; https://imagej.nih.gov/ij/).

### 4.9. RNA Interference and Transfections

The siRNAs are shown in Appendix A. All siRNAs were synthesized by and purchased from Hokkaido System Science (Sapporo, Japan). AllStars Negative Control siRNA (Qiagen. Tokyo, Japan) was used as a negative control. All siRNA transfections were carried out using Lipofectamine RNAiMAX (Thermo Fisher Scientific, Waltham, MA, USA).

### 4.10. Quantitative Real-Time PCR (RT-PCR)

We used the semi-quantitative PCR method to detect mRNAs in human ovarian cancer cell lines and on the ovarian surface epithelium. Human ovarian surface epithelial cell total RNA was purchased from Cosmo Bio. Total RNA was extracted from cell lines with the RNeasy Mini Kit (Qiagen), and subjected to qPCR. cDNA was synthesized from 1 µg of total RNA using a High-Capacity cDNA Reverse Transcription Kit (Applied Biosystems) with an oligo-dT primer. Fast SYBER Green Master Mix (Applied Biosystems) was used for amplification and the samples were run on the Step One system (Applied Biosystems). The primers were designed to detect spliced mRNA, and their sequences are shown in Appendix A. All primers were synthesized by and purchased from Hokkaido System Science.

### 4.11. Peri-Mesothelial Cell Migration Assay

HPMCs were isolated, as we previously reported [58], from the tumor-free human omentum of patients with malignant ovarian tumors. The HPMCs were cultured on collagen-coated plates in RPMI-1640 media supplemented with 10% FBS and penicillin/streptomycin until confluency. Confluent monolayers of HPMCs were incubated, and then GFP-labeled ES-2 cells with control or MITF-targeted siRNA were seeded. The time-lapse images were acquired every 10 min for 600 min with a BZX-800 microscope and then analyzed with BZX-800 View Analyzer Software. The average migration velocities and mean squared displacements (MSDs) of individual cancer cells were obtained.

### 4.12. Proteomic Analysis

Samples were prepared according to our previous report [59]. A two-fold change was set as the value for the geographic average between the paired groups as determined by the Mascot program (Matrix Science, Tokyo, Japan), as previously described [60]. The database for annotation, visualization, and integrated discovery (http://david.abcc.ncifcrf.gov, version 6.8), a bioinformatics web tool, was employed to analyze the enrichment of biological pathways using the Kyoto Encyclopedia of Genes and Genomes with all the altered proteins. STRING, another bioinformatics web tool (http://www.string-db.org/, version 10), was used to perform an interactome analysis. Additionally, Proteomaps with altered proteins were generated to visualize the differential contribution of biological pathways (http://bionic-vis.biologie.uni-greifswald.de/, version 2.0).

### 4.13. Analysis of Cell Surface Markers

Expression of cell surface MCAM was examined by flow cytometry. Briefly, 10^6^ cells were stained with phycoerythrin- or Alexa Fluor 647-conjugated mouse anti-CD146 monoclonal antibody (BD Biosciences) for 30 min on ice, washed three times with 1% BSA in PBS, and then suspended with 1% BSA in PBS prior to analysis on a FACS Aria II (BD Biosciences).

## Figures and Tables

**Figure 1 ijms-21-09776-f001:**
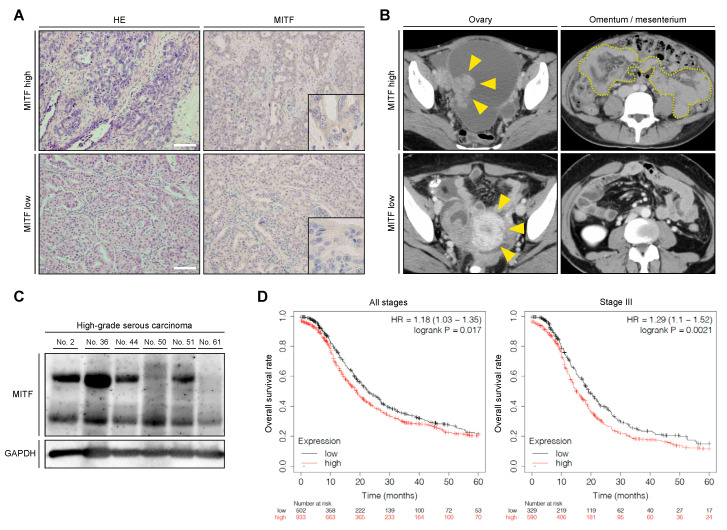
Clinical impact of microphthalmia-associated transcription factor (MITF) expression in ovarian cancer (OvCa) tissue. (**A**) Representative image of the H&E staining and immunohistochemistry of the MITF at a primary OvCa site. Scale bar: 100 μm. (**B**) Representative image of computed tomography in patients with high and low MITF expression in OvCa tissue. Arrowheads indicate the primary OvCa site and the enclosed area highlights peritonitis carcinomatosis. (**C**) Immunoblot analysis of the tissue samples extracted from primary OvCa. (**D**) Kaplan–Meier curves of progression-free survival in all patients and those with stage III OvCa, stratified by differences in MITF expression.

**Figure 2 ijms-21-09776-f002:**
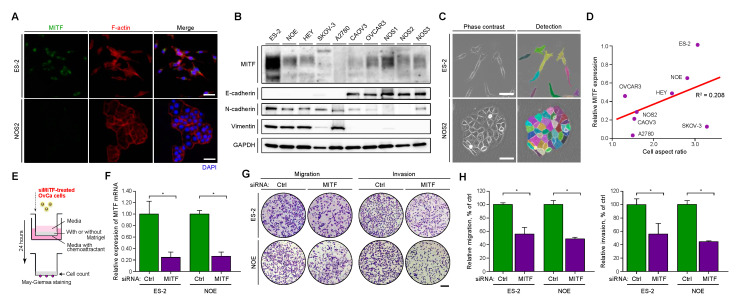
MITF expression is associated with motility and invasion of OvCa cells. (**A**) Representative images of immunofluorescence detecting MITF (left panels) and F-actin (middle panels) in ES-2 and NOS2 cells. Scale bar: 50 μm. (**B**) Immunoblots of MITF, E-cadherin, N-cadherin, vimentin, and GAPDH with proteins extracted from OvCa cell lines. (**C**) Images of the cell aspect ratios in OvCa cells. Scale bar: 50 μm. (**D**) Correlation of MITF expression levels with the immunoblot results and cell aspect ratio of each OvCa cell line. (**E**) Schematic protocol for the motility and invasion assay. (**F**) Relative expression levels of MITF mRNA in ES-2 and NOE cells under siRNA treatment. * *p* < 0.05 (*t*-test). (**G**) Representative images in the motility and invasion assays. Scale bar: 200 μm. (**H**) Relative migration and invasion percentages in ES-2 and NOE cells under siRNA treatment. * *p* < 0.05 (*t*-test).

**Figure 3 ijms-21-09776-f003:**
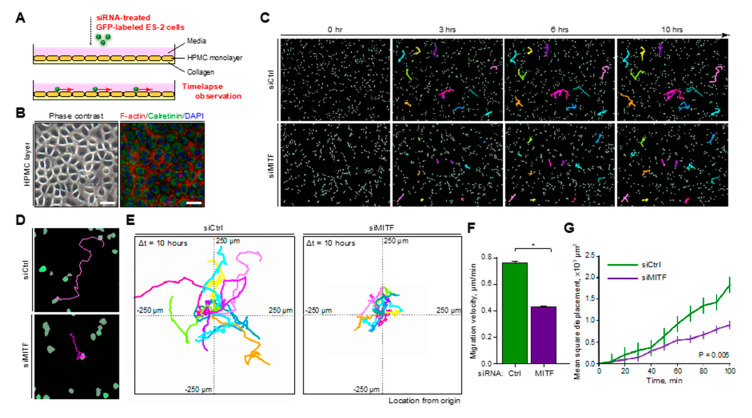
MITF is associated with peri-mesothelial migration of OvCa cells. (**A**) Schematic protocol of the peri-mesothelial migration assay. (**B**) Representative phase-contrast and immunofluorescence images of an HPMC monolayer. Scale bar: 50 μm. (**C**) Time-lapse images and movement locus of randomly selected GFP-labeled ES-2 cells treated with control or MITF-targeted siRNA. (**D**) Representative images of the traced movement locus of GFP-labeled ES-2 cells treated with control or MITF-targeted siRNA. (**E**) Integrated traced movement loci of ES-2 cells in each treatment group. (**F**,**G**) Average migration velocity and chronological mean square displacement of GFP-labeled ES-2 cells treated with control or MITF-targeted siRNA. * *p* < 0.05 (*t*-test).

**Figure 4 ijms-21-09776-f004:**
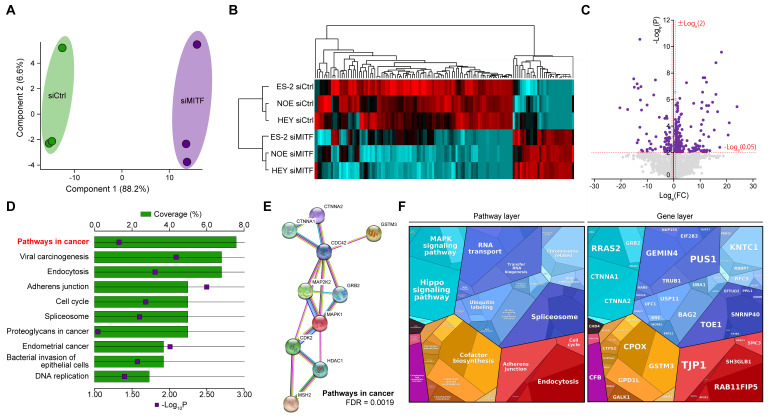
Proteomic analysis reveals the relationship between MITF and cancer pathways. (**A**) A principal component analysis (PCA) map illustrating similarities in altered protein expression between each sample group. (**B**) A heat map illustrating differentially expressed proteins in ES-2, NOE, and HEY cells treated with control or MITF-targeted siRNA. (**C**) A volcano plot indicating the fold changes in expression and *p* values for all proteins identified in ES-2 cells treated with control or MITF-targeted siRNA. (**D**) Pathway enrichment analysis and interactomes created with proteins related to cancer pathways. (**F**) Proteomaps depicting the fold changes and associated functions of altered proteins within OvCa cells treated with MITF-targeted siRNA compared to those treated with control siRNA.

**Figure 5 ijms-21-09776-f005:**
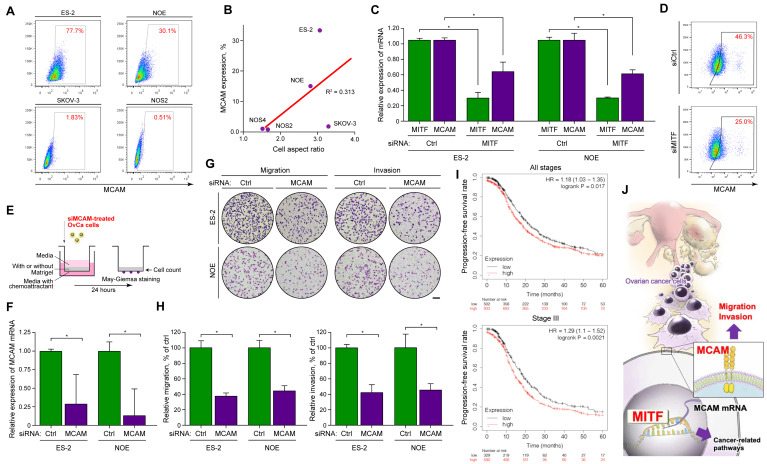
MCAM modulated by MITF promotes motility and invasion of OvCa cells. (**A**) Flow cytometry images demonstrating MCAM on OvCa cells. (**B**) Correlation between MCAM expression on flow cytometry and cell aspect ratios of OvCa cell lines. (**C**) Relative expression of MITF and MCAM mRNA in ES-2 and NOE cells treated with control or MITF-targeted siRNA. * *p* < 0.05 (*t*-test). (**D**) Flow cytometry images of MCAM on ES-2 cells treated with control or MITF-targeted siRNA. (**E**) Schematic protocol for the motility and invasion assay. (**F**) Relative expression of MCAM mRNA treated with control or MCAM-targeted siRNA. (**G**) Representative images from the motility and invasion assays. Scale bar: 200 μm. (**H**) Relative migration and invasion percentages of ES-2 and NOE cells under siRNA treatment. * *p* < 0.05 (*t*-test). (**I**) Kaplan–Meier curves of all patients and those with stage III OvCa, stratified by differences in MCAM expression. (**J**) Graphical abstract of the findings in this study.

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
