# Peer review of "Microphthalmia-Associated Transcription Factor-Dependent Melanoma Cell Adhesion Molecule Activation Promotes Peritoneal Metastasis of Ovarian Cancer"

_ijms, 2020, doi:10.3390/ijms21249776_

Round 1

Reviewer 1 Report

The paper written by Kitami, Yoshiara and their colleagues have well demonstrated the importance of MITF (microphthalmia-associated transcription factor) in association with peritoneal metastasis of ovarian cancer. They suggest that MITF/MCAM axis causes metastasis of ovarian cancer, thus providing a possibility of using MITF as a biomarker that can predict patient prognosis. Also, the authors are well aware of the limitations of their study, as mentioned in the discussion section.

Overall, the paper is well written with a logical flow. However, there are some major revisions they have to make before finally publishing the paper.

Major comments:

  1. The authors have well shown that MITF-high cells (ES-2) have high migratory and invasion characteristics (figures 2 and 3). However, the authors should also consider using siMITF in MITF-low cells (such as SKOV-3) and consider performing the same set of experiments using such cell lines to conclude that MITF is associated with metastasis.
  2. The authors have well incorporated proteomic profiling data into their manuscript to explain the relationship of MITF and cancer signaling pathways (figure 4). However, the authors should explain this data clearly in the results section instead of briefly mentioning the figures (line 152).
  3. The relationship between MITF and MCAM expression has been well shown in ES-2 and NOE cell lines that carry high levels of MITF. However, the authors should also show the association between MITF and MCAM levels in MITF-low cells to conclusively state the possibility of their association in OVCA cells (figure 5 and lines 246-247).
  4. The discussion section should be reorganized and emphasize the merit of your study in detail.

Minor Comments:

  1. Please carefully go through the manuscript again before making final submissions, as there are multiple typos and flaws in the manuscript.
  • Lines 98 to 104 is redundant (previous sentences say exact same idea)
  • Figure labeling in the manuscript must be re-checked (e.g. Lines 113, 132)
  1. In figure 2D and 5B, please add labeling of cell names to each dots to clarify the association between cell aspect ratio and relative MITF expression.

Author Response

Comments and Suggestions for Authors The paper written by Kitami, Yoshiara and their colleagues have well demonstrated the importance of MITF (microphthalmia-associated transcription factor) in association with peritoneal metastasis of ovarian cancer. They suggest that MITF/MCAM axis causes metastasis of ovarian cancer, thus providing a possibility of using MITF as a biomarker that can predict patient prognosis. Also, the authors are well aware of the limitations of their study, as mentioned in the discussion section. Overall, the paper is well written with a logical flow. However, there are some major revisions they have to make before finally publishing the paper. Major comments: 1. The authors have well shown that MITF-high cells (ES-2) have high migratory and invasion characteristics (figures 2 and 3). However, the authors should also consider using siMITF in MITF-low cells (such as SKOV-3) and consider performing the same set of experiments using such cell lines to conclude that MITF is associated with metastasis.  [Author response] Thank you for your suggestion. We analyzed whether MITF contributes to the migration and invasion abilities of MITF-low cells by transfecting the SKOV-3 cell line, a MITF-low cell line, with siRNA against MITF. As a result, there was no significant change in either migration or invasion abilities under MITF down-regulated conditions. This may be due to the fact that SKOV-3 originally has low migration and invasion ability and low expression of MITF, so the effect of knockdown is small. These results were shown as Supplementary Figure 1 and described in the text (lanes 114-115).   2. The authors have well incorporated proteomic profiling data into their manuscript to explain the relationship of MITF and cancer signaling pathways (figure 4). However, the authors should explain this data clearly in the results section instead of briefly mentioning the figures (line 152). [Author response] Thank you for your kind indication. Proteome analysis showed that MITF is also involved in cancer malignancy and cancer-related pathways, including the MAPK pathway, in ovarian cancer. It has also been reported that MITF binds directly to the promoter region of MCAM and is involved in cancer invasion. With this in mind, we investigated the relationship between MITF and MCAM.   3. The relationship between MITF and MCAM expression has been well shown in ES-2 and NOE cell lines that carry high levels of MITF. However, the authors should also show the association between MITF and MCAM levels in MITF-low cells to conclusively state the possibility of their association in OVCA cells (figure 5 and lines 246-247).  [Author response] Thank you for your kind indication. We analyzed the expression of MCAM on MITF-low cells, which are SKOV-3 and NOS2 cells, using flow cytometry analysis. The expression of MCAM on these cell lines was low (Figure 5A), and their migration and invasion abilities are also low (Supplementary figures 1 and 2, lanes 115-117). MCAM expressions of two cell lines with low MITF expression, NOS2 and SKOV3, are shown in Figure 5B using flow cytometry analysis.  And, as it is the mesenchymal subtype that has been reported to cause peritoneal dissemination and poor prognosis in ovarian cancer, we selected mesenchymal-like OvCa cells, ES-2 and NOE to determine whether MITF and MCAM are involved in cancer progression.   4. The discussion section should be reorganized and emphasize the merit of your study in detail.  [Author response] Thank you for your kind indication. In our study, we found that MITF promotes metastasis of ovarian cancer by regulating the expression of MCAM, which in turn promotes invasion and migration. Therefore, we believe that MITF and MCAM have the potential to be prognostic biomarkers and therapeutic targets, as shown in Figure 1D and 5I. In addition, MITF inhibitors have already been developed, and anti-MCAM antibodies have been reported to be effective as therapeutic agents for MCAM-positive tumors, including in vivo, so there is much potential for clinical application of MITF/MCAM-axis targeted therapy. We believe this is the merit of our study and have described it emphatically (lanes 257-263).   Minor Comments: 1. Please carefully go through the manuscript again before making final submissions, as there are multiple typos and flaws in the manuscript.  [Author response]  Line 154: The English wording was changed from "the two groups" to "each group" because it was incorrect. Line 224: We used an abbreviation without explanation, so we corrected from ECM to extracellular matrix. Line 255: We have corrected the verb from "decreased" to "increased". • Lines 98 to 104 is redundant (previous sentences say exact same idea) [Author response]  According to the reviewer’s direction, these redundant sentences have been removed. • Figure labeling in the manuscript must be re-checked (e.g. Lines 113, 132) [Author response] Thank you for pointing that out. Line 114: We have corrected the cell line name on Figure 2G from HEY to NOE.    Furthermore, Figure 3G and 3H were errors in Figure 2G and 2H have been corrected. Line136: Figure 4C was an error in Figure 3C and has been corrected. 2. In figure 2D and 5B, please add labeling of cell names to each dots to clarify the association between cell aspect ratio and relative MITF expression. [Author response] According to the reviewer’s direction, we added labeling of cell names to each dot.      We sincerely thank you for your helpful assistance. We hope that our revisions meet with your requirements.

Reviewer 2 Report

This study by Kitami K et al, suggests that MITF modulates the melanoma cell adhesion molecule (MCAM/CD146), which promotes peritoneal dissemination. Based on that, MITF could be a novel molecular target that potentially predicts prognosis in patients with ovarian cancer. The manuscript is straightforward, well written, and concise, and has clear results. Definitely deserves to be published and is a valuable contribution to the “International Journal of Molecular Sciences”. Some minor flaws need to be addressed before publication.

Minor points:

[1] “Introduction”, Page 1/14, Lines 39-40:

Peritoneal dissemination, which is one of the most common types of metastasis in the abdominal cavity, is frequently observed in patients with advanced OvCa [4,5].”.

At that point, please do report that there is also a separate subset of patients with serous peritoneal papillary carcinoma, managed similarly to stage III/IV ovarian cancer. The recommended treatment integrates aggressive cytoreductive surgery, hyperthermic intraperitoneal chemotherapy, and systemic chemotherapy to remove the macroscopic tumor, eradicate the microscopic residual disease, and control the microscopic metastasis.

Recommended reference: Rassy E, et al. Narrative review on serous primary peritoneal carcinoma of unknown primary site: four questions to be answered. Ann Transl Med 2020. doi: 10.21037/atm-20-941.

[2] “Introduction”, Page 1/14, Lines 40-42:

Even when primary tumors are optimally resected, recurrent tumors frequently emerge in the peritoneum despite the use of conventional platinum-based chemotherapies [6,7]”.

From the therapeutic point of view, it has also been supported that neoadjuvant treatment is non-inferior to the standard primary debulking strategy in the management of patients who are fit for either procedure. Please, do mention that the estimated median PFS is approximately 12–18 months.

Recommended reference: Boussios S, et al. Poly (ADP-Ribose) Polymerase Inhibitors: Talazoparib in Ovarian Cancer and Beyond. Drugs R D. 2020;20:55-73.

[3] “Discussion”, Page 8/14, Lines 248-250:

Treatment of siRNA against MITF reduced MCAM mRNA levels as well as MCAM expression on the cell surface (Figure 5C and D), suggesting that MITF directly or indirectly regulates MCAM expression”.

There is evidence for miRNA with demonstrated efficacy in decreasing the metastasis of tumor cells. Furthermore, they can also target epithelialmesenchymal transition with potential clinical applications. Such miRNA are potential candidates for translational medicine in ovarian cancer treatment.

Recommended reference: Palma Flores C, et al. MicroRNAs driving invasion and metastasis in ovarian cancer: Opportunities for translational medicine (Review). Int J Oncol. 2017 May;50(5):1461-1476.

[4] A workflow diagram for the study would be of benefit for the readers.

Author Response

Comments and Suggestions for Authors
This study by Kitami K et al, suggests that MITF modulates the melanoma cell adhesion molecule (MCAM/CD146), which promotes peritoneal dissemination. Based on that, MITF could be a novel molecular target that potentially predicts prognosis in patients with ovarian cancer. The manuscript is straightforward, well written, and concise, and has clear results. Definitely deserves to be published and is a valuable contribution to the “International Journal of Molecular Sciences”. Some minor flaws need to be addressed before publication.

Minor points:

[1] “Introduction”, Page 1/14, Lines 39-40:

“Peritoneal dissemination, which is one of the most common types of metastasis in the abdominal cavity, is frequently observed in patients with advanced OvCa [4,5].”.
At that point, please do report that there is also a separate subset of patients with serous peritoneal papillary carcinoma, managed similarly to stage III/IV ovarian cancer. The recommended treatment integrates aggressive cytoreductive surgery, hyperthermic intraperitoneal chemotherapy, and systemic chemotherapy to remove the macroscopic tumor, eradicate the microscopic residual disease, and control the microscopic metastasis.

Recommended reference: Rassy E, et al. Narrative review on serous primary peritoneal carcinoma of unknown primary site: four questions to be answered. Ann Transl Med 2020. doi: 10.21037/atm-20-941.
[Author response]
Thank you for your kind indication. We followed your instructions and added a description of recommended treatments for advanced ovarian cancer and primary peritoneal carcinoma. Hyperthermic chemotherapy is not a standard treatment in Japan, so it was not included in the recommended treatments. We have added the recommended article (line 40-46).

[2] “Introduction”, Page 1/14, Lines 40-42:

“Even when primary tumors are optimally resected, recurrent tumors frequently emerge in the peritoneum despite the use of conventional platinum-based chemotherapies [6,7]”.

From the therapeutic point of view, it has also been supported that neoadjuvant treatment is non-inferior to the standard primary debulking strategy in the management of patients who are fit for either procedure. Please, do mention that the estimated median PFS is approximately 12–18 months.

Recommended reference: Boussios S, et al. Poly (ADP-Ribose) Polymerase Inhibitors: Talazoparib in Ovarian Cancer and Beyond. Drugs R D. 2020;20:55-73.
[Author response]
Thank you for your kind indication. Following your suggestion, we added the description of PFS and the recommended reference (line 46).

[3] “Discussion”, Page 8/14, Lines 248-250: 

“Treatment of siRNA against MITF reduced MCAM mRNA levels as well as MCAM expression on the cell surface (Figure 5C and D), suggesting that MITF directly or indirectly regulates MCAM expression”.

There is evidence for miRNA with demonstrated efficacy in decreasing the metastasis of tumor cells. Furthermore, they can also target epithelial–mesenchymal transition with potential clinical applications. Such miRNA are potential candidates for translational medicine in ovarian cancer treatment.

Recommended reference: Palma Flores C, et al. MicroRNAs driving invasion and metastasis in ovarian cancer: Opportunities for translational medicine (Review). Int J Oncol. 2017 May;50(5):1461-1476.
[Author response]
Thank you for your kind indication. As you pointed out, there is evidence of therapeutic applications using microRNAs, and there are actually reports of microRNAs targeting MCAM, so we added this description and recommended reference (line 260-263).

[4] A workflow diagram for the study would be of benefit for the readers.
[Author response]
Thank you for your kind suggestion. We have shown schematic protocol of each experiment and graphical abstract in the final figure, which seems at least adequate to demonstrate the flow of this research. We hope your understanding.

We sincerely thank you for your helpful assistance. We hope that our revisions meet with your requirements.

Round 2

Reviewer 1 Report

The authors have well explained and further edited their manuscript to suit the revision comments. They provided solid evidence, using Supplementary files, to support their explanations for the manuscript and their experiments. Also, they have clarified the merit of this study in the discussion section, and now it is suitable for being published in IJMS.